# Biological Control of *Verticillium* Wilt and Growth Promotion in Tomato by Rhizospheric Soil-Derived *Bacillus amyloliquefaciens* Oj-2.16

**DOI:** 10.3390/pathogens12010037

**Published:** 2022-12-26

**Authors:** Dongli Pei, Qingchen Zhang, Xiaoqin Zhu, Lei Zhang

**Affiliations:** 1Henan Provincial Engineering Research Center for Development and Application of Characteristic Microorganism Resources, College of Biology and Food, Shangqiu Normal University, Shangqiu 476000, China; 2Institute of Crops Molecular Breeding, Henan Academy of Agricultural Sciences, Zhengzhou 450002, China

**Keywords:** *Bacillus amyloliquefaciens*, *Verticillium dahliae*, *Verticillium* wilt, biocontrol, growth promoting

## Abstract

*Verticillium* wilt disease caused by *Verticillium dahliae* seriously affects tomato quality and yield. In this work, strain Oj-2.16 was isolated from rhizosphere soil of the medicinal plant *Ophiopogon japonicas* and identified as *Bacillus amyloliquefaciens* on the basis of morphological, physiological, and biochemical characteristics and 16S rDNA sequencing. Strain Oj-2.16 exhibited a high inhibition rate against *V. dahliae*, and the hyphae inhibited by Oj-2.16 were found to be destroyed on scanning electron microscopy. Lipopeptide and dipeptide genes were detected in the Oj-2.16 genome by PCR amplification involved in surfactin, iturin, fengycin, and bacilysin biosynthesis. In pot experiments, the biocontrol efficacy of strain Oj-2.16 against *Verticillium* wilt in tomato was 89.26%, which was slightly higher than the efficacy of the chemical fungicide carbendazim. Strain Oj-2.16 can produce indole acetic acid, siderophores, assimilate various carbon sources, and significantly promoted the growth of tomato seedlings by increasing plant height, root length, stem width, fresh weight, and dry weight by 44.44%, 122.22%, 80.19%, 57.65%, 64.00%, respectively. Furthermore, defense-related antioxidant CAT, SOD, POD, and PAL enzyme activities significantly increased and MDA contents significantly decreased in tomato seedlings treated with strain Oj-2.16 upon inoculation of *V. dahliae* compared with the pathogen-inoculated control. In summary, we concluded that *B. amyloliquefaciens* Oj-2.16 could be used as a promising candidate for the biocontrol of *Verticillium* wilt and as plant growth stimulator of tomato.

## 1. Introduction

During cultivation, tomato plants could be infected by several pathogens, such as *Verticillium dahliae* (*Vd*), *Fusarium oxysporum*, *Phytophthora infestans*, *Botrytis cinerea*, and so forth. *Verticillium* wilt is a highly destructive soil-borne fungal disease caused by *V. dahliae* [1]. The disease has changed from secondary disease to main disease in China in recent years. Heavily infected tomato plants have small fruits or no fruits, which significantly reduces tomato quality and yield, especially in a protected field. Though the *Ve* resistance gene against *Verticillium* wilt is widespread in the current tomato varieties, it confers resistance to race 1 strains of *V. dahliae* and not to race 2 strains [1]. At present, there are no commercially available cultivars that are resistant to *Verticillium* wilt. Use of chemical pesticides is the main approach to control *Verticillium* wilt [2], but chemical control harms human health and the environment, and the control effects decrease gradually due to the drug resistance of pathogens. Therefore, biocontrol of *Verticillium* wilt by antagonistic microorganisms has been a promising and sustainable measure because of less environment pollution and a lower health danger compared with the chemical control.

Members of the genus *Bacillus* are ubiquitous nonpathogenic bacteria in nature. *Bacillus amyloliquefaciens* can be used as a biocontrol agent for a number of pathogens and as a plant growth–promoting bacterium, because it does not harm humans and animals and is eco-friendly [3]. It produces a series of metabolites inhibiting fungal and bacterial activities, with the advantages of fast growth, strong resistance, and simple nutrition requirements. *B. amyloliquefaciens* is reported as a most promising bacterium in many kinds of plants to biocontrol lots of bacterial and fungal plant pathogens, including *Colletotrichum orbiculare* [4], *F. oxysporum* [5], *Sclerotinia sclerotiorum* [6], *Xanthomonas citri* [7], *Pectobacterium carotovorum* [8], *B. cinerea* [9], and *Rhizopus oryzae* [10]. However, few studies reported a biocontrol effect and prevention mechanism of *B. amyloliquefaciens* on *Verticillium* wilt in tomato plants.

In this study, strain Oj-2.16 was isolated from rhizosphere soil of the medicinal plant *Ophiopogon japonicus* and identified by morphological characteristics and 16S rDNA sequencing. Its biocontrol effect on tomato *Verticillium* wilt and growth-promoting effect on tomato seedlings were assayed, and the activities of the tomato defense-related enzyme induced by strain Oj-2.16 were explored.

## 2. Materials and Methods

### 2.1. Materials

Fungal pathogens were provided by the key laboratory of plant–microbe interactions of Henan province, China, including *V. dahliae*, *Botryosphaeria dothidea*, *F. graminearum*, *F. verticillioides*, and *F. oxysporum*. Carbendazim was purchased from Sichuan Guoguang Agrochemical Co., Ltd.

### 2.2. Isolation and Purification of Rhizospheric Soil Bacteria of O. japonicus

Five samples of rhizosphere soil were obtained from an *O. japonicas* field in Shangqiu, Henan Province, China. Bacteria were isolated from *O. japonicas* rhizosphere soil attached to roots according to a previous report [11]. The single colonies of bacteria were picked and purified by streak culture on LB agar, and the strains were stored in a tube containing 30% (*v*/*v*) glycerol.

### 2.3. Determination of Bacterial Antagonistic Activity

The panel confrontation way was used to screen the antagonistic activity of rhizosphere soil bacteria to different kinds of fungal plant pathogens. The 0.5 cm diameter fungus cake was set at the PDA plate center and cultivated for 24 h at 28 °C in the incubator. Bacterial strains were cultivated overnight at 37 °C in LB medium and placed to three symmetrical sites around the fungus cake on the PDA plate and with a 2.5 cm distance to the plate edge. Bacterial strains were not inoculated to the control plate. All treatments were followed by a culture in the incubator at 28 °C. The diameters of the inhibited pathogen and the control pathogen were measured after 4 days incubation. The relative inhibition rates were calculated using the formula: (diameter of the pathogen on the control plate—diameter of the pathogen on the treated plate/diameter of the pathogen on the control plate) × 100%.

### 2.4. Scanning Electron Microscopy Observations of the Inhibited Hyphae of Pathogenic Fungi

The mycelial morphology and structural change of the pathogen in antagonistic treatment and control treatment were observed by scanning electron microscopy (Hitachi SU8010, Hitachi Higher Technologies Corp., Tokyo, Japan) as described previously [12].

### 2.5. Morphological, Physiological, and Biochemical Characteristics of Strain Oj-2.16

The color and colony appearance were detected on the macromorphological level, whilst the bacterial structure was observed under a microscope. The physiological and biochemical indices were detected in terms of Bergey’s Manual of Determinative Bacteriology, including Gram staining, methyl red reaction, Voges–Proskauer reaction, gelatin liquefaction, nitrate reduction, D-glucose utilization, fructose utilization, sucrose utilization, mannitol utilization, and starch hydrolysis [13]. IAA production was tested with the Salkowski method [14]. Siderophore production was detected on the medium with Chrome Azurol S blue [15].

### 2.6. 16S rDNA Sequence Identification and Phylogenetic Analyses

The genomic DNA of the bacterial strain Oj-2.16 was obtained by the rapid DNA isolation kit of the bacterial genomic (Sangon Biotech Co., Ltd., Shanghai, China) on the basis of the manufacturer’s instructions. The 16S rDNA sequences of strain Oj-2.16 were amplified, adopting the forward primer fD1 (5′-AGAGTTTGATCCTGGCTCAG-3′) and reverse primer rP1 (5′-TACCTTGTTACGACTT-3′) by PCR. The reaction was performed using 25 μL 2×Taq mixture, 5 μL DNA template, 2.5 μL forward and reverse primers, and 15 μL ddH_2_O. The PCR reaction conditions were 94 °C for 3 min; followed by 35 cycles of 94 °C for 30 s, annealing at 55 °C for 30 s, and extending at 72 °C for 1 min; and finally extending at 72 °C for 10 min. The PCR fragments were visualized using 1% DNA agarose gel electrophoresis. The amplified fragments were sequenced by Sangon Biotech Co., Ltd. (Shanghai, China). The obtained sequence was then submitted to GenBank. The homology analysis of 16S rDNA sequences was performed by the BLAST tool [16]. Finally, the phylogenetic tree was built to analyze the taxonomic status of the strain by the MEGA using the neighbor joining method [17].

### 2.7. PCR Amplification of Lipopeptide and Dipeptide Biosynthetic Genes

DNA was amplified by primers for lipopeptide genes, including surfactin, fengycin, iturin, and dipeptide gene bacilysin (Table 1). The PCR mix was the same as in Section 2.6, and the PCR reaction conditions were 94 °C for 5 min; followed by 32 cycles of 94 °C for 15 s, annealing at 55 °C for 15 s, and extending at 72 °C for 30 s/kb; and finally, extending at 72 °C for 10 min [18]. The PCR fragments were visualized by 1% DNA agarose gel electrophoresis.

### 2.8. Biocontrol and Growth-Promoting Effects of the Strain Oj-2.16 on Tomato in Pot Experiments

Seeds of the tomato cultivar Moneymaker were picked and sterilized as follows: dipping in 3% NaClO for 5 min and washing three times using aseptic water. After treatment, the seeds were germinated in culture dishes with wet aseptic filter papers at 26 °C with 16 h light/8 h dark cycle per day. The germinating seedlings were planted in a pot (11 cm high, 9.5 cm in diameter) with autoclaved soil (2:1 nutritional soil: vermiculite). The pots were put in growth chamber at 26 °C with 16 h light/8 h dark. After four true leaves were formed, each pot of tomato seedlings was managed according to the following: (1) treatment by 20 mL of aseptic water; (2) treatment by 20 mL suspensions of strain Oj-2.16 (1.0 × 10^8^ CFU mL^−1^); (3) first treatment by 20 mL of aseptic water and two days later, treatment by 20 mL spore suspensions (1.0 × 10^8^ spores mL^−1^) of *V. dahliae*; (4) first treatment by 20 mL suspensions of strain Oj-2.16 and two days later, treatment by 20 mL spore suspensions of *V. dahliae*; (5) first treatment by 20 mL carbendazim (1 g L^−1^) and two days later, treatment by 20 mL spore suspensions of *V. dahliae*. One treatment was performed on 20 plants, each plant in one pot. Five treatments with 100 plants were placed randomly. The experiment was conducted in three independent repetitions. Statistical analyses were conducted using the Duncan’s test with SPSS V21.0.

After 20 days of *V. dahliae* spore inoculation, disease incidence, disease index, and biocontrol efficacy were detected as described by Chen et al. [19]. Disease incidence (%) = [(diseased plants number)/total number of plants treated)] × 100; disease index (%) = {[Σ(number of plants in each disease grade × disease grade)]/(total number of plants × the highest grade)} × 100; biocontrol efficacy (%) = [(the control disease index—the treatment disease index)/the control disease index] × 100. The disease severity of *Verticillium* wilt in tomato plants was categorized into five grades (grade 0–4) as the criteria of disease grades presented by Wu et al. [20], the percentage of infected tomato leaves was categorized as 0, ≤25, 26–50, 51–75, ≥76%.

The plant growth status was then assessed. The tomato plants were rooted out, and they were rinsed using running water and dried naturally. The shoot height, root length, stem width, and fresh weight of each plant were tested. The tomato plants were oven-dried for 3 days at 80 °C to test the dry weight.

### 2.9. Antioxidant Enzyme Activity Analysis

After 6 days of *V. dahliae* inoculation, antioxidant enzyme activity levels of each treatment were measured. First, 1 g of tomato leaves were picked, washed, dried, put into a precooled mortar, mixed using 8.0 mL of 0.05 mol L^−1^ (pH 7.8) phosphate buffer, grinded into homogenate, and centrifuged at 10,000 rpm for 15 min and 4 °C by centrifuge (Eppendorf 5804R, Eppendorf SE, Hamburg, Germany). The supernatant was obtained, which was the crude enzyme extracted from tomato leaves.

The activity of superoxide dismutase (SOD) was tested by the nitro blue tetrazolium (NBT) photoreduction method [21], and the amount of enzyme which was required to inhibit 50% of the photoreduction of NBT was defined as one unit of SOD. The activity of peroxidase (POD) was tested by the guaiacol colorimetry method [21], and the amount of enzyme which was required to increase the A_470_ value by 0.01 per minute was identified as one unit of POD. The activity of catalase (CAT) was measured by the hydrogen peroxide method [21], and the amount of enzyme which was required to reduce the A_240_ value by 0.1 per minute was defined as one unit of CAT. Phenylalanine ammonia lyase (PAL) was measured by the phenylalanine method [21], and the amount of enzyme which was required to increase the A_290_ value by 0.01 per minute was defined as one unit of PAL. The malondialdehyde (MDA) concentration was measured according to the thiobarbituric acid method [22].

## 3. Results

### 3.1. Isolation of Bacteria and Screening of Antagonistic Strains against Pathogenic Fungi

A total amount of 69 bacterial strains were acquired from the rhizosphere soil of *O. japonicus*. The antagonism of 69 bacterial strains to five different species of pathogenic fungi was screened using the panel confrontation method, resulting in 21 strains with an inhibitory effect on *F. oxysporum*, eight strains with an effect on *V. dahliae*, eleven strains with an effect on *B. dothidea*, nine strains with an effect on *F. verticillioides*, and six strains with an effect on *F. graminearum*. The strains with highest mycelial inhibition activity were re-screened, and five strains were selected with a higher inhibition rate from 52% to 71% against five species of pathogenic fungi, and named Oj-2.3, Oj-2.10, Oj-2.16, Oj-2.20, and Oj-2.23. Among the five strains, the strain Oj-2.16 had the highest inhibition rate against *V. dahliae* (67.50%; Figure 1) and was selected for further experiments.

### 3.2. Scanning Electron Microscopy Observation of Inhibited Hyphae of Pathogen

The scanning electron microscopy observation indicated that the hyphae of *V. dahliae* were straight and evenly thick in the control group (Figure 2A), while the hyphae of *V. dahliae* inhibited by strain Oj-2.16 in panel confrontation were found to be dehydrated, distorted, dissociated, and forming hollows (Figure 2B–D).

### 3.3. Physiological and Biochemical Characteristics of Strain Oj-2.16

After 72-h culturing of strain Oj-2.16, the colony edge was irregular, milky white, opaque, dry, dull, and easily spread on the plates. The results of physiological and biochemical characteristics of strain Oj-2.16 showed that Gram staining, Voges–Proskauer reaction, nitrate reduction, IAA production, siderophore production, starch hydrolysis, D-glucose utilization, fructose utilization, and mannitol utilization were positive, while gelatin liquefaction, methyl red reaction, and sucrose utilization were negative (Table 2).

According to biochemical characterization, strain Oj-2.16 could be *B. amyloliquefaciens*.

### 3.4. 16S rDNA Sequencing and Phylogenetic Analysis of Strain Oj-2.16

The genomic DNA of strain Oj-2.16 was isolated and amplified with PCR. The size of 16s rDNA amplicon was 1421 bp (GenBank accession number MT830870). The BLAST results showed that strain Oj-2.16 had 99.79% sequence identity with *B. amyloliquefaciens* (FN597644) [23]; both were clustered into one branch by phylogenetic tree construction (Figure 3). On the basis of physiological and biochemical properties and 16s rDNA sequence analysis, strain Oj-2.16 was identified as *B. amyloliquefaciens*.

### 3.5. PCR Amplification of the Lipopeptide and Dipeptide Biosynthetic Genes from Strain Oj-2.16

PCR amplification from the genomic DNA of strain Oj-2.16 indicated the presence of three lipopeptide biosynthetic genes, including fengycin, surfactin, iturin, and one gene of the non-ribosomally synthesized dipeptide antibiotic bacilysin (Figure 4). So, strain Oj-2.16 has the potential of inhibiting plant pathogens by producing different kinds of antifungal compounds.

### 3.6. Biological Efficacy of B. amyloliquefaciens Oj-2.16 on Verticillium wilt in Pot Experiments

Over time, tomato plant leaves exhibited symptoms with disease spot from a faint yellow color to tawny color, and plants withered in cases of severe disease in the *Vd* inoculation treatment groups. *B. amyloliquefaciens* Oj-2.16 reduced wilt severity of tomato seedlings inoculated with *Vd*. The disease incidence of *Verticillium* wilt in tomato plants inoculated with *B. amyloliquefaciens* Oj-2.16 decreased by 66.67%, the disease index decreased by 62.50 compared to only *Vd* inoculation, and the biocontrol efficacy reached 89.26%. The biocontrol efficacy of *B. amyloliquefaciens* Oj-2.16 was higher than the efficacy of carbendazim treatment, but with no significant difference between the two treatments (Table 3, Figure 5).

### 3.7. Growth-Promoting Effect of B. amyloliquefaciens Oj-2.16 on Tomato Seedlings

There was significant increase in plant height, root length, stem width, fresh weight, and dry weight in inoculation *B. amyloliquefaciens* Oj-2.16 treatment groups compared to non-treated tomato seedlings (Table 4, Figure 5). The increases of plant height, root length, stem width, fresh weight, and dry weight were 44.44%, 122.22%, 80.19%, 57.65%, 64.00%, respectively. There was also significant increase in plant height, root length, stem width, fresh weight, and dry weight in *B. amyloliquefaciens* Oj-2.16 and the *Vd* inoculation treatment group compared to the *Vd* inoculation treatment group. The *Vd* inoculation seedlings treated with carbendazim were also observed, the results showed that *B. amyloliquefaciens* Oj-2.16 and *Vd* inoculation treatment were significantly higher in plant height, root length, stem width, fresh weight, and dry weight than the carbendazim and *Vd* treatment.

### 3.8. Effects of B. amyloliquefaciens Oj-2.16 on Antioxidant Enzyme Activities and MDA Content of Tomato Seedlings

After 6 days of *V. dahliae* inoculation, antioxidant SOD, POD, CAT, and PAL enzyme activities and MDA content of tomato seedlings in different treatments were detected (Table 5). Tomato seedlings of *B. amyloliquefaciens* Oj-2.16 treatment expressed significantly higher antioxidant enzyme activities of SOD, POD, CAT, and PAL compared to non-treated control seedlings. *Vd* inoculation treatment also significantly increased all these enzyme activities compared to the control without *Vd* inoculation. *Vd* inoculation and *B. amyloliquefaciens* Oj-2.16 treatment showed the highest activities and was significantly higher for all these enzymes (SOD, POD, CAT and PAL) compared to *Vd* inoculation alone. The *Vd* inoculation and carbendazim treatment expressed significantly higher activity of CAT and PAL, but significantly less activities of SOD and POD than *Vd* inoculation alone. MDA content in *Vd* inoculation treatment was significantly higher than in the control, and compared with the value under *Vd* inoculation alone, MDA content under *Vd* inoculation plus *B. amyloliquefaciens* Oj-2.16 was significantly lower.

## 4. Discussion

Biological control has been recognized as a promising and sustainable measure because of less pollution to the environment and a lower health danger compared with the chemical control. *B. amyloliquefaciens* with a higher antagonistic activity against phytopathogens can be successfully used as a potential biocontrol agent to control diverse crop diseases, including potato scab caused by *Streptomyces griseoplanus* [24]; wheat root rot caused by *Bipolaris sorokiniana* [25]; ginseng gray mold caused by *B. cinerea* [26]; alfalfa anthracnose caused by *C. truncatum* [27]; tomato bacterial wilt caused by *Ralstonia solanacearum* [28]; tomato bacterial canker caused by *Clavibacter michiganensis* [29]; wheat *Fusarium* head blight caused by *F. graminearum* [30]; and sugar beet leaf spot disease caused by *Pseudomonas syringae* pv. aptata [31]. It was also reported that *B. amyloliquefaciens* strains can promote the growth of plants by producing IAA hormones, siderophores, volatile organic compounds (VOCs) and improving the usability of nutrients [32,33,34].

The present study revealed that *B. amyloliquefaciens* Oj-2.16 suppressed the growth of the fungal pathogen *V. dahliae* and controlled *Verticillium* wilt in tomato. The in vitro inhibition rate of hyphal growth was 67.50% and the biocontrol efficacy of *B. amyloliquefaciens* Oj-2.16 was 89.26%, which was slightly higher than the efficacy of the chemical fungicide carbendazim in pot experiments. *B. amyloliquefaciens* is one of the most prevalently studied biocontrol agents. However, only limited experiments have been performed to biocontrol *Verticillium* wilt by *Bacillus*. Similar results were obtained by the utilization of *B. atrophaeus* to impart resistance to *Verticillium* wilt of cotton [35] and *B. velezensis* to control *Verticillium* wilt of olive trees [36].

In addition, inoculation with *B. amyloliquefaciens* Oj-2.16 significantly increased shoot height, root length, stem width, fresh weight, and dry weight of tomato seedlings compared with the control, attributed to the features of strain Oj-2.16 to produce IAA, siderophores, assimilate various carbon sources, including glucose, fructose, and mannitol. These results showed that the antagonistic strain Oj-2.16 could also serve as a plant growth stimulator or biofertilizer, which could be superior to the chemical fungicide carbendazim that does not have the ability of promoting growth. Ji et al. [31] reported that *B. amyloliquefaciens* Ba13 significantly increased tomato plant height and fresh weight by 10.98% and 20.15%, respectively, which were lower than those increased by *B. amyloliquefaciens* Oj-2.16, namely 44.44% and 57.65%, respectively.

*B. amyloliquefaciens* is well known for its efficacy in the biocontrol of plant diseases, which involves mechanisms like secretion of antimicrobial substances, competition for nutrients and ecological niche, induction of host systemic resistance (ISR), growth promotion, and enhancement of colonization ability [37,38]. Induction of the enzyme activities related to plant defense is a form of systemic resistance, which plays key roles in disease resistance response of plants [39,40]. Our results showed that *B. amyloliquefaciens* Oj-2.16 triggered tomato plant resistance to *Verticillium* wilt by increasing the activities of defense-related antioxidant enzymes. POD, SOD, CAT, and PAL enzyme activities significantly increased in the plants treated with strain Oj-2.16 upon inoculation of *V. dahliae* compared with the pathogen-inoculated control. The enhanced activities of antioxidant enzymes can effectively scavenge harmful reactive oxygen species to maintain the stability of plant defense systems, so the treated tomato returned to normal growth (Figure 5). MDA content is the index of the level of membrane-lipid peroxidation. Compared with such value under the mere *Vd* pathogen-inoculated control, MDA content under *V. dahliae* inoculation and *B. amyloliquefaciens* Oj-2.16 treatment was significantly lower, indicating a lower damage degree of the membrane system.

Scanning electron microscopy observation revealed that the hyphae of *V. dahliae* were destroyed to be dehydrated, distorted, dissociated and forming hollows, after fungi were treated with *B. amyloliquefaciens* Oj-2.16 spore suspension in panel confrontation experiments. *B. amyloliquefaciens* might secrete some antimicrobial substances, such as antimicrobial proteins and peptides, which can damage the mycelial structure, including deformation, expanding hyphae, and leaking protoplasm [26,41]. Genes involved in antifungal lipopeptide and dipeptide biosynthesis were detected in *B. amyloliquefaciens* strain Oj-2.16, suggesting that Oj-2.16 could produce fengycin, surfactin, iturin, and bacilysin. To explore its biocontrol mechanism and broaden its application, the expression, isolation, and function identification of peptide products produced by strain Oj-2.16 in the process of controlling tomato *Verticillium* wilt need to be studied further.

In summary, *B. amyloliquefaciens* Oj-2.16 was firstly isolated from rhizosphere soil of *O. japonicas* with a broad antagonistic activity against *V. dahliae*, *B. dothidea*, *F. oxysporum*, *F. graminearum*, and *F. moniliforme*. *B. amyloliquefaciens* Oj-2.16 can effectively biocontrol tomato *Verticillium* wilt disease and promote tomato growth. Strain Oj-2.16 can produce IAA, siderophores, assimilate various carbon sources including glucose, fructose, and mannitol, all of the features contributed to promoting growth. *B. amyloliquefaciens* Oj-2.16 triggered tomato plant resistance to *Verticillium* wilt by increasing the levels of defense-related antioxidant enzymes POD, SOD, CAT, and PAL. Induction of the activities of plant defense enzymes is a form of systemic resistance, which is the main antifungal mechanism to suppress tomato *Verticillium* wilt disease.

## Figures and Tables

**Figure 1 pathogens-12-00037-f001:**
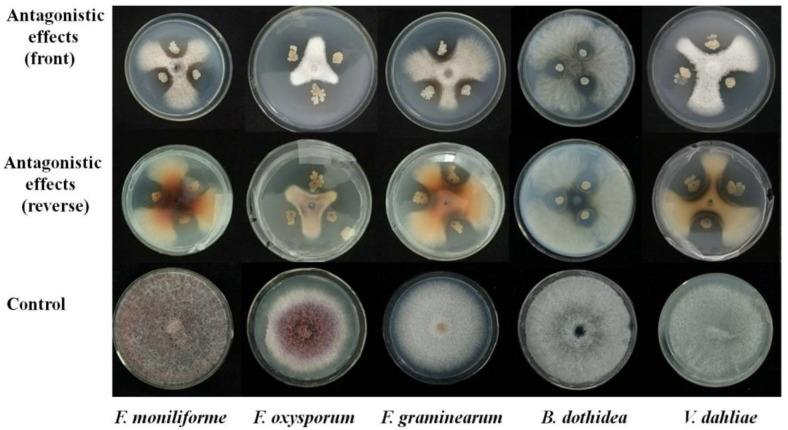
Antagonistic effect of strain Oj-2.16 on the growth of five species of fungal pathogens.

**Figure 2 pathogens-12-00037-f002:**
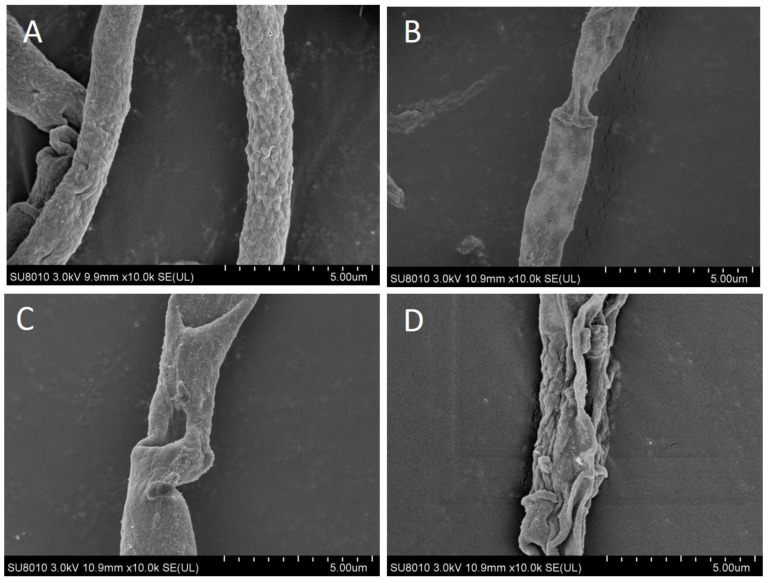
Scanning electron microscopic photography of *Verticillium dahliae* hyphae. (**A**) The untreated hyphae of *V. dahliae*; (**B**–**D**) the hyphae of *V. dahliae* treated with strain Oj-2.16.

**Figure 3 pathogens-12-00037-f003:**
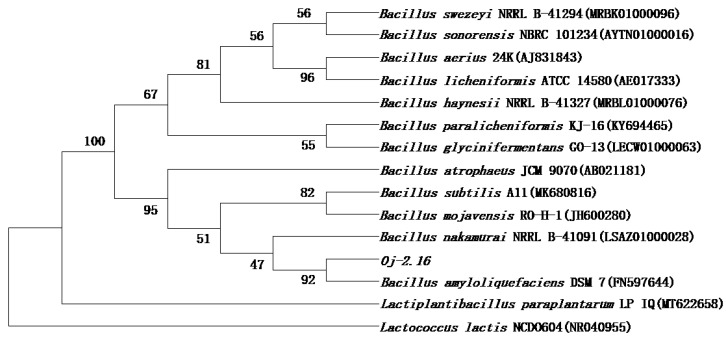
A neighbor-joining phylogenetic tree construction of strain Oj-2.16.

**Figure 4 pathogens-12-00037-f004:**
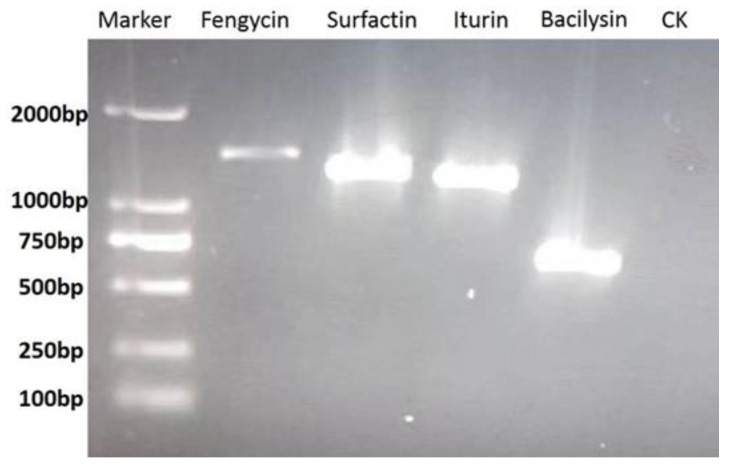
PCR amplification of genes coding for lipopeptides and dipeptides of strain Oj-2.16. CK, positive control (without DNA).

**Figure 5 pathogens-12-00037-f005:**
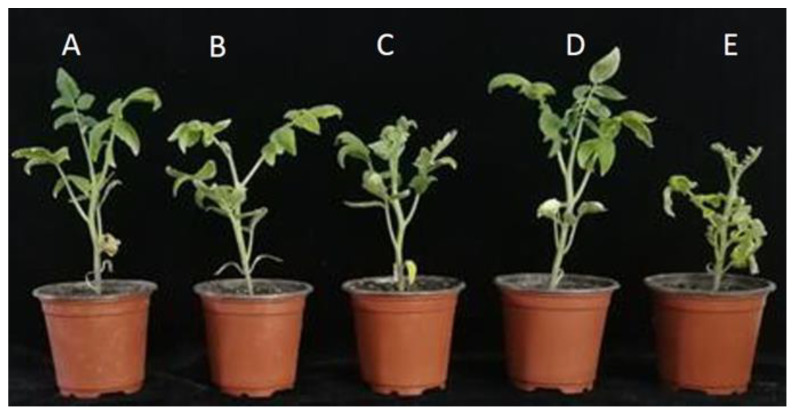
Effects of strain Oj-2.16 against *Verticillium* wilt disease and on growth promotion of tomato seedlings. (**A**) Inoculation of *Vd* and strain Oj-2.16; (**B**) *Vd* inoculation and carbendazim treatment; (**C**) aseptic distilled water; (**D**) only strain Oj-2.16 inoculation; (**E**) only *Vd* inoculation.

**Table 1 pathogens-12-00037-t001:** Primers to amplify biosynthetic genes of lipopeptide and dipeptide [18].

Lipopeptides and Dipeptides	Annealing Temperature	Size (Base Paires)	Primers
Surfacin	56.3	1298	5′ CGGTGATCTTGCGAAGCTTTAT 3′ 5′ CGCTTTCGTTCTGCCATTCT 3′
Fengycin	56.3	1447	5′ CGGCCATTCGCTCATCTTTTAT 3′ 5′ GTTTCCGCTTCATCAGTCTCTTC 3′
Iturin	55.9	1244	5′ ACCTCACCTTGATCGGCTATAC 3′ 5′ TGGTGGGCGAAAGAAGTTTATG 3′
Bacilysin	52.8	657	5′ TCATGACTCTTTCGCCTCT 3′ 5′ GAATGGGATAACGGAGTAAGAC3′

**Table 2 pathogens-12-00037-t002:** Physiological and biochemical characteristics of the strain Oj-2.16.

Physiological and Biochemical Characteristics	Oj-2.16
gelatin liquefaction	−
methyl red reaction	−
Voges-Proskauer reaction	+
nitrate reduction	+
D-glucose utilization	+
fructose utilization	+
sucrose utilization	−
mannitol utilization	+
starch hydrolysis	+
Gram staining	+
IAA production	+
siderophore production	+

“+” indicates positive, “−” indicates negative.

**Table 3 pathogens-12-00037-t003:** Biocontrol efficacy of *B. amyloliquefaciens* Oj-2.16 against *V. dahliae* in tomato pot experiments.

Treatment	Disease Incidence (%)	Disease Index	Biocontrol Efficacy (%)
*Vd* + Oj-2.16	30.00 ± 5.00 c	7.50 ± 1.25 c	89.26 ± 2.05 a
*Vd* + Carbendazim	41.67 ± 7.64 b	11.67 ± 1.91 b	83.27 ± 3.21 a
*Vd*	96.67 ± 2.89 a	70.00 ± 2.50 a	— —

Note: The data in the table are means ± SD; The different letters within columns are significantly different at *p* < 0.05 level by Duncan’s test.

**Table 4 pathogens-12-00037-t004:** Effect of *Bacillus amyloliquefaciens* Oj-2.16 on the biomass of tomato plants.

Treatments	Plant Height (cm)	Root Length (cm)	Stem Width (mm)	Fresh Weight (g)	Dry Weight (g)
Oj-2.16	16.64 ± 1.26 a	8.60 ± 0.70 a	5.73 ± 0.60 a	4.02 ± 0.55 a	0.41 ± 0.06 a
*Vd* + Oj-2.16	15.98 ± 1.51 a	6.72 ± 0.71 b	5.03 ± 0.54 b	3.36 ± 0.35 b	0.33 ± 0.04 b
*Vd* + Carbendazim	12.43 ± 1.35 b	4.55 ± 0.67 c	4.09 ± 0.42 c	2.58 ± 0.38 c	0.27 ± 0.03 c
CK	11.52 ± 1.13 b	3.87 ± 0.52 c	3.18 ± 0.45 d	2.55 ± 0.40 c	0.25 ± 0.03 c
*Vd*	9.04 ± 0.79 c	3.02 ± 0.38 d	2.40 ± 0.49 e	1.79 ± 0.40 d	0.19 ± 0.04 d

Note: The data in the table are means ± SD; The different letters within columns are significantly different at *p* < 0.05 level by Duncan’s test.

**Table 5 pathogens-12-00037-t005:** Antioxidant enzyme activities and MDA content of tomato leaves under different treatments.

Treatments	SOD (U/mg Protein)	POD (U/mg Protein)	CAT (U/mg Protein)	PAL (U/mg Protein)	MDA (µmol·g^−1^ FW)
*Vd*+Oj-2.16	7.24 ± 0.27 a	17.86 ± 0.43 a	28.58 ± 0.77 a	85.76 ± 1.18 a	8.16 ± 0.53 c
*Vd*	6.28 ± 0.31 b	15.77 ± 0.40 b	18.79 ± 0.40 c	60.77 ± 1.16 c	14.74 ± 0.60 a
*Vd*+Carbendazim	4.41 ± 0.37 d	14.29 ± 0.40 c	24.65 ± 0.65 b	72.27 ± 0.97 b	10.21 ± 0.49 b
Oj-2.16	5.39 ± 0.44 c	12.08 ± 0.31 d	15.89 ± 0.61 d	43.46 ± 0.78 d	5.25 ± 0.59 d
CK	3.57 ± 0.29 e	8.81 ± 0.30 e	10.12 ± 0.43 e	22.52 ± 0.58 e	4.36 ± 0.46 d

Note: The data in the table are means ± SD; The different letters within columns are significantly different at *p* < 0.05 level by Duncan’s test.

## Data Availability

Not applicable.

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
