# Peer review of "Biological Control of Verticillium Wilt and Growth Promotion in Tomato by Rhizospheric Soil-Derived Bacillus amyloliquefaciens Oj-2.16"

_pathogens, 2022, doi:10.3390/pathogens12010037_

Round 1
Reviewer 1 Report
Data for Oj-2.16 activity against Verticillium wilt in tomato look promising.
Introduction:
Explain what is meant by "especially in a protected field" when referring to yield and quality loss due to Verticillium wilt in tomato.
Please include a reference for "Use of chemical pesticides is the main approach to controlling Verticillium wilt." There are few chemical pesticides registered specifically for Verticillium wilt.
Materials and Methods:
Please include the pot size so the treatment volumes can be taken into context of soil/treatment ratio.
Please explain and justify the use of carbendazim as a chemical treatment comparison. Why was this chosen as a treatment? Include if it is registered for use on tomato for management of Verticillium wilt.
Please provide (brief) additional detail about the experimental design. Each treatment was performed on 20 plants in triplicate. Were the pots placed in 3 blocks with plant/treatment randomized in each block, was it completely random, or were three experiments conducted in tandem? (as example of the detail needed to determine if statistical analysis is appropriate). Whether blocked or conducted at different time points, an analysis of variance with treatment/block or time/and if 20 plants were individual or analyzed as sampling error would be helpful.
Indicate if the experiments were conducted in glasshouse or growth chamber.
Results:
Several of the biomass parameters (Table 4) show no difference between carbendazim treatment check; and biocontrol efficacy (Table 3) is not different between carbendazim and Oj-2.16 so again it is important to explain if carbendazim is actually a commonly used, effective, registered treatment for Verticillium wilt in tomato.
Discussion:
Especially if the Discussion section begins "Compared with chemical control..." and second paragraph has "...which was to be superior to chemical fungicide carbendazim in pot experiments." It is important to define the effectiveness of carbendazim in this context. The data are good without the chemical treatment, but if it is used, more detail must be provided to justify its use as a treatment here.
Check all the spellings of dahliae. Unfortunately, dahlia is a beautiful flower and word processing often autocorrects dahliae to dahlia. In checking references used for this article, some were published with at least one incorrect spelling of dahlia instead of dahliae.
Reviewer 2 Report
The article Biological control of Verticillium wilt and growth promotion in tomato by rhizospheric soil-derived Bacillus amyloliquefaciens Oj-2.16, by Pei et al. provides useful information on the antagonistic activity, both in vitro and in vivo, of the bacterium against the phytopathogenic fungus Verticillium dahliae, in tomato plants. Genetic and biochemical analyzes characterize the activity of the antagonistic bacterium, while biocontrol assays in pots, in addition to highlighting a significant reduction in the disease, show an increase in some vegetative parameters of the plants treated with the bacterium.
The following revisions are recommended.
Line 27, Keywords: species names in italics, Verticillium dahliae (correct also lines 31, 33, 130, 132, 133, 135, etc.);
line 36, Ve?;
lines 55-56: delete sentence;
line 58: Its biocontrolling, eliminates space;
line 65: V.;
lines 126-127: provide some information on soil type;
paragraph 3.4: blasting Oj-2.16 16s rDNA sequence results 100% identity with B. siamensis KCTC 13613 16s rDNA (KY643639) and B. velezensis CR-502 16s rDNA (ON597433) sequences. Why you didn't include these species in your phylogenetic tree? And why you compare your 16s rDNA sequence with the whole genomic sequence of B. amyloliquefaciens DMS7?
Reviewer 3 Report
The study by Pei et al. evaluated the biological control of Verticillium wilt by Bacillus amyloliquefaciens. Exploring alternative measures to control different plant pathogens is of immense importance in light of the finding substitutes for harmful chemical pesticides. The topic of the study is significant, and various methods were used to characterize the in vitro antifungal, in vivo biocontrol, plant growth activity and potential to induce the production of antioxidant enzymes in tomato plants. Also, antifungal activity of B. amyloliquefaciens was detected by scanning electron microscopy. There are some minor suggestions listed in the comments below. The major concern is the time of Verticillium dahliae application in the biocontrol experiment, and a comment is provided below. Also, the discussion needs to be broadened with existing literature data.
Line 25: change “a plant growth” to “as plant growth”
Line 31: change “Tomato is infected by several pathogens during cultivation” to “During cultivation tomato could be infected by several pathogens”
Line 53: provide references for studies related to the control effect and mechanism of B. amyloliquefaciens on Verticillium wilt in tomato. Also, what makes your study exceptional compared to existing studies?
Line 55: rephrase the sentence to “One of the most promising biocontrol methods against plant pathogens is to screen antagonistic bacteria from plant rhizospheric soil”
Line 64: add “fungal” before “pathogens”
Line 65: add country
Line 76: add “fungal” before “plant pathogen”
Line 92: Rephrase sentence “The color and morphology of bacterial strain Oj-2.16 was observed by eyes, and bacterial structure was observed under a microscope” to “ The color and colony appearance were detected on the macromorphological level, whilst the bacterial structure was observed under a microscope”
Line 99: check subsection number
Line 114: bacylisin isn’t a lipopeptide yet a non-ribosomally synthesized dipeptide antibiotic, please change in Table 1. Accordingly, change the subsection and Table 1 title.
Line 115: change “reaction system” to “mix”
Line 119: melting or annealing temperature?
Line 129: does control for V. dahlia set 2 days after other controls were applied? It is crucial for comparing control results to the treatment considering that in treatment 4 you first applied an antagonist and then after two days pathogen was applied. If that was not the case, then the results were not comparable.
Line 173: change “The strains with obvious inhibitory activity” to “The strains with highest mycelial inhibition activity”
Line 175: change “against five kinds” to “against five species”
Line 177: end sentence with “and was selected for further experiments”
Line 178: remove the sentence
Line 184: change “dehydration” to “dehydrated” and move “forming hollows” to the end of the sentence
Line 191: change “provoked” to some other phrase like “it was easily spread on the plates”
Line 195: according to biochemical characterization what you concluded about the identification of analyzed species
Line 198: check subsection number
Line 200: do you submit the sequence to GenBank or the accession number MT830870 was related to the reference strain?
Line 209: change “four” to “three”
Line 210: add “non-ribosomally synthesized dipeptide antibiotic” before bacylisin. Also, change bacyllisin to bacylisin, and check spell throughout manuscript.
Line 212: considering that you analyzed Bacillus strain for the antifungal activity please change “antibacterial substance” to “antifungal compounds”
Line 214: change “lipopeptides genes” to “genes coding for lipopeptides and bacylisin”
Line 222: check fungicide spelling
Line 283: there were no statistically significant differences between the application of strain B. amyloliquefaciens Oj-2.16 and carbendazim, so it cannot be stated that “B. amyloliquefaciens Oj-2.16 was 89.26%, which was to be superior to chemical fungicide carbendazim in pot experiments”
Line 285: add some antifungal mechanisms of action by B. amyloliquefaciens, discuss
Line 293: discuss the role of B. amyloliquefaciens as a plant growth-promoter from other studies, compare to your results
Line 310: change “mycelial structures” to “hyphae”
Line 311: change “dehydratation” to “dehydrated”
Line 316: change bacylisin to a dipeptide compound according to suggestions in the methods and results section
Line 317: Conclusion needs to be broadened, shortly conclude identified activities of B. amyloliquefaciens Oj-2.16
